# Electron heating and thermal relaxation of gold nanorods revealed by two-dimensional electronic spectroscopy

Aude Lietard[1,2], Cho-Shuen Hsieh[1,2], Hanju Rhee[3] & Minhaeng Cho [1,2]

To elucidate the complex interplay between the size and shape of gold nanorods and their electronic, photothermal, and optical properties for molecular imaging, photothermal therapy, and optoelectronic devices, it is a prerequisite to characterize ultrafast electron dynamics in gold nanorods. Time-resolved transient absorption (TA) studies of plasmonic electrons in various nanostructures have revealed the time scales for electron heating, lattice vibrational excitation, and phonon relaxation processes in condensed phases. However, because linear spectroscopic and time-resolved TA signals are vulnerable to inhomogeneous line-broadening, pure dephasing and direct electron heating effects are difficult to observe. Here we show that femtosecond two-dimensional electronic spectroscopy, with its unprecedented time resolution and phase sensitivity, can be used to collect direct experimental evidence for ultrafast electron heating, anomalously strong coherent and transient electronic plasmonic responses, and homogenous dephasing processes resulting from electron-vibration couplings even for polydisperse gold nanorods.

[1] Center for Molecular Spectroscopy and Dynamics, Institute for Basic Science (IBS), Seoul 02841, Republic of Korea. [2] Department of Chemistry, Korea University, Seoul 02841, Republic of Korea. [3] Seoul Center, Korea Basic Science Institute (KBSI), Seoul 02841, Republic of Korea. Correspondence and requests for materials should be addressed to H.R. (email: hjrhee@kbsi.re.kr) or to M.C. (email: mcho@korea.ac.kr)

Gold nanorods (AuNRs) have been used in a variety of biological and biomedical applications[1] because the collective and coherent oscillations of their conduction band electrons, referred to as surface plasmon resonance (SPR) modes, induced by an incident electric field are tunable by changing the aspect (length-to-width) ratio $R$. The modes along the long and short axes of AuNRs, which correspond to longitudinal and transverse SPR (LgSPR and TrSPR) excitations, respectively, appear in the long- and short-wavelength regions in the absorption spectrum[1–4], with the LgSPR frequency strongly depending on $R$. Typically, synthesized AuNRs have a broad distribution of aspect ratios, which gives rise to the inhomogeneous broadening of the LgSPR band. This limits their use in optical sensing applications[5] because inhomogeneous line broadening can average out the desired optical sensing effects of the intrinsic SPR bands of individual nanorods.

Although transmission electron microscopy is of great use in determining the size (and aspect ratio) heterogeneity of polydisperse nanorods, sample preparation is difficult and there is a lack of reproducibility due to the limited number of images taken for statistical analysis. Another much simpler optical spectroscopic method is to fit the experimentally measured absorption spectrum to a collection of calculated homogeneously linebroadened spectra for nanorods with different aspect ratios[6]. Attempts have also been made to characterize the transient optical response of a single AuNR or nanoantenna, and the ultrafast plasmonic dynamics of an individual metallic nanometer object has been elucidated using pump-probe spectroscopy and microscopy[7,8]. However, it is still difficult and often impossible to clearly distinguish between the contributions of homogeneous and inhomogeneous line broadenings for an ensemble of inhomogeneously distributed gold nanorods by merely examining one-dimensional (in frequency) steady-state or pump-probe transient absorption spectra[2,3,7,9–17].

Two-dimensional electronic spectroscopy (2DES)[18–24] has often been used to investigate the electronic coupling and dynamics of multi-chromophore systems by analyzing spectrally resolved off-diagonal signals on their 2DES spectra. Another important observable that can be extracted from a 2DES spectrum is the slope of nodal line that separates positive and negative 2D peaks. The nodal line slope (NLS) has been shown to be of exceptional use in gaining information on the degree of dynamic inhomogeneity of solvated molecular systems in condensed phases[19,24].

Here we demonstrate that the 2DES, based on four-wave mixing technique, is capable of measuring nonlinear optical signal fields for LgSPR-excited AuNRs and provides direct information on the homogeneous dynamics of electron heating, electron–electron (e–e), and electron–phonon (e–ph) scatterings as well as on the inhomogeneous distribution of AuNRs. Furthermore, we show that spectral interference patterns observed in negative-time 2DES spectra are related to nonlinear transient grating processes of AuNRs that intrinsically cannot be observed with conventional frequency-resolved pump-probe measurement methods.

## Results

**Pump-probe TA spectroscopy.** One of the most popular methods that is useful to measure time constants of various relaxation processes after photo-excitation of SPR modes of nanoparticles is TA spectroscopy. We measured time-resolved TA ($\Delta A$) spectra with respect to pump-probe delay (waiting) time $T_w$ up to 700 200 for ps, which is long enough to cover the entire range of photothermal dynamics for AuNRs[2,11,16]. In our TA and 2DES measurements, the center wavelengths of the pump and probe are the same and their bandwidths are narrower than that of the AuNR LgSPR band (Fig. 1a) so that a subset of AuNRs is selectively excited and probed. To cover the entire LgSPR band and to

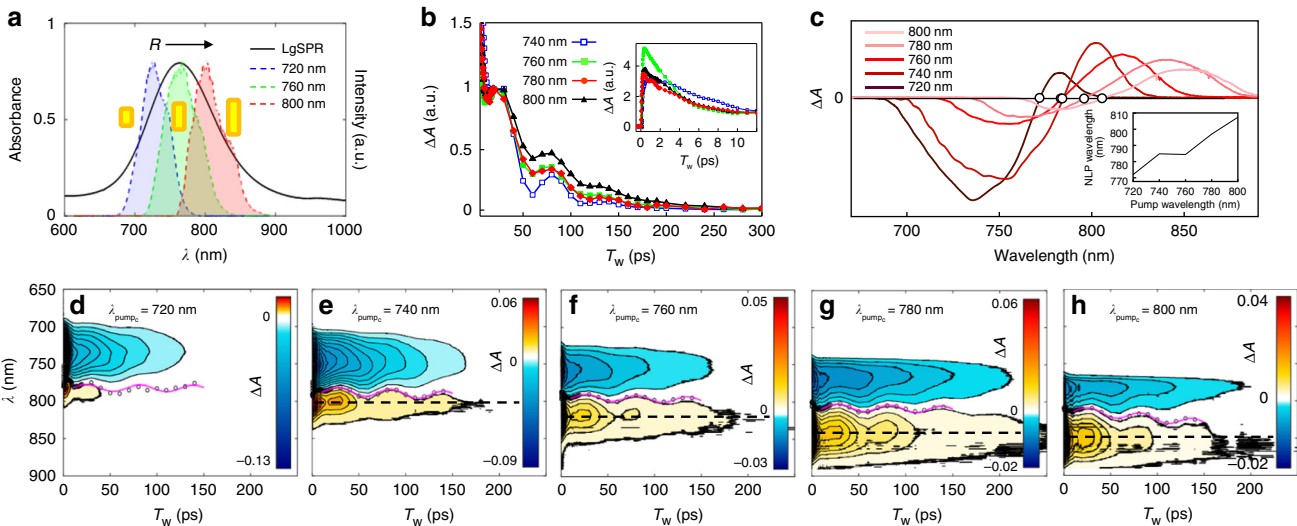

**Fig. 1** LgSPR absorption spectrum, and time and frequency-resolved transient absorption spectra. **a** The LgSPR absorption band (black solid line) of AuNR sample and the laser spectra of different center wavelengths (color-shaded areas) used in the present work. The aspect ratio ($R$) of resonant AuNRs increases with the LgSPR absorption wavelength. **b** Time profiles of the positive TA signals (taken at the black dashed lines in **e–h**) are plotted with respect to pump-probe delay time. The center wavelengths of both the pump and probe pulses are tuned from 740 to 800 nm. Each TA time profile is appropriately scaled for easier comparison. Inset: Enlarged view of the TA time profiles at short times up to 12 ps. **c** The TA spectra taken at 1 ps of pump-probe delay time ($T_w$) for different pump-probe wavelengths. Inset: plot of nodal point (white circles) wavelength of the TA spectra versus pump wavelength. **d–h** Temporal evolutions of the TA spectra of AuNRs up to 250 ps after excitation with pump at center wavelengths of $\lambda_{pump_C} = 720$ (**d**), 740 (**e**), 760 (**f**), 780 (**g**), and 800 nm (**h**). The open circles represent nodal points (zero-crossing points) across which the sign of the TA signal is inverted and the pink lines are their fits, which clearly show a slow anti-correlated oscillation in amplitude due to the coherently excited extensional vibration of the nanorods

examine the pump center wavelength dependence of relaxation dynamics for different subsets of AuNRs, the TA spectra were measured at various pump-probe center wavelengths from 720 to 800 200 for nm (Fig. 1). Unlike previous studies that have used high-energy pump photons whose wavelengths were much shorter than those of the two SPR modes[7,12,13,15,16], we exclusively excite LgSPR modes with pump photons whose energy is much lower than either the interband or TrSPR mode transition energy.

For all pump-probe wavelengths, a negative peak on the blue side and a positive peak on the red side appear in the TA spectra (Fig. 1d–h), despite their relative amplitudes varying with pump center wavelength, $\lambda_{pump_c}$. On the basis of a scenario in which the dielectric constant changes as a result of the electron heating induced by femtosecond photoexcitation, this TA spectral feature arises from both spectral broadening and the red-shift of each homogeneous LgSPR band[1,2,7,13,25]. The relative amplitude of the positive peak to the negative peak in our TA spectra decreases with the pump center wavelength ($\lambda_{pump_c}$) and even becomes close to zero at $\lambda_{pump_c} = 720$ nm (Fig. 1d). The disappearance of the positive TA peak when excited on the blue side of the LgSPR band primarily originates from the limited bandwidth of the probe pulse, which does not fully cover the red side of the LgSPR band, where the positive peak should arise due to a frequency red-shift of the entire inhomogeneous LgSPR band upon photo-excitation of the AuNRs.

Another plausible explanation for the weaker positive signal on the blue side is that the significant spectral overlap of the adjacent intrinsic (homogeneous) LgSPR bands from AuNRs with different aspect ratios can lead to a cancellation of those overlapped positive and negative TA signals on the blue side of the probe wavelength, which will be shown in the simulation results below (Fig. 3g). Indeed, the fact that the nodal points of the pump-probe TA spectra do not remain constant but red-shift with the pump-probe center wavelength (Fig. 1c–h) indicates that even our AuNR sample with relatively low polydispersity still has considerable spectral inhomogeneity over the entire LgSPR band. Nonetheless, it should be emphasized that one cannot investigate the wavelength-dependence of the photo-excited spectral signatures and dynamics of the LgSPR band within the bandwidth of the laser pulse used (Fig. 1a) with the conventional pump-probe TA spectroscopy because its signal is not free from sample heterogeneity. We shall thus show that this critical limitation of linear and TA measurement methods can be overcome by 2DES.

Figure 1b plots the decay profiles of the positive TA signals taken at the dashed lines in Fig. 1e–h, which can be described by considering the following three major contributions: two exponentially decaying components and one damped oscillation component. The time constants for the fast and slow exponentially decaying components are 2–5 ps and 50–80 ps, respectively, which correspond to the e–ph and ph–ph relaxations, respectively[9,11,13,16] (see Supplementary Table 1 for the summarized time constants of the photothermal relaxations measured at all the pump wavelengths). However, these time constants do not strongly depend on the pump center wavelength nor the aspect ratio of the AuNRs. This is consistent with previous findings by the El-Sayed group[1]. The slowly oscillating component can be attributed to a coherent excitation of extensional vibrations along the long axis of each AuNR, which can be induced by instantaneous heating after the photoexcitation of AuNRs[16]. It should be noted that the nodal line between the positive and negative TA signals (Fig. 1d–h) exhibits far more pronounced anti-correlated amplitude oscillations than the conventional TA time profiles shown in Fig. 1b. Thus, it becomes clear that the time-resolved and dispersed pump-probe spectroscopy enables

one to obtain a more accurate fit to the damped oscillation component in the decaying TA signal.

According to the previous TA study on interband transition of AuNRs, the oscillation period of the extensional vibration along the long axis of a nanorod depends only on the length of the nanorod and increases linearly with this parameter[16]. For AuNRs with large size inhomogeneity (high polydispersity), their different aspect ratios and the degree of polydispersity in sample can be differentiated by the oscillation period ($T_{osc}$) of the extensional vibration and its damping time ($t_{damp}$) obtained from the corresponding TA signal[16]. In the present case, however, both $T_{osc}$ (~56 ps) and $t_{damp}$ (~50 ps) do not strongly depend on the pump center wavelength (Supplementary Table 1) responsible for excitation of subensembles of AuNRs with different aspect ratios (Fig. 1a). This is because the AuNR sample studied here has a relatively low polydispersity (about 10% of the lengths range from 35 to 41 nm) and therefore it is difficult to resolve them only by linear spectroscopic or time-domain TA measurement methods. In the following section, we show that the 2DES can resolve inhomogeneously broadened features of AuNRs even for the same sample, and is thus of use to judge the degree of polydispersity in a AuNR solution sample with a fairly narrow size distribution.

**Two-dimensional electronic spectroscopy of AuNR.** We present our experimental setup (scheme) for the Fourier transform (FT) 2DES based on two-beam geometry in Fig. 2a[26]. A single femtosecond laser pulse is split into two; one is converted into coherent twin pump pulses ($E_1$ and $E_2$) with a pulse shaper, and the other is used as a probe pulse ($E_3$). There are two controllable delay times for the 2DES pulse sequence, coherence ($\tau$) and waiting ($T_w$) times. The coherence time $\tau$ between the two pump pulses ($E_1$ and $E_2$) propagating collinearly is controlled by the pulse shaper and the waiting time $T_w$ between the second pump ($E_2$) and the probe ($E_3$) pulses is scanned with a motorized delay stage.

Figure 2b–d illustrate how a 2DES spectrum, $S(\omega,T_w,\omega')$, of the LgSPR band of AuNR at a given $T_w$ is obtained by controlling the $\tau$-scan of the twin pump pulses. Each femtosecond pump pulse, $E_1(\omega)$ and $E_2(\omega)$, in frequency domain can be decomposed into constituting electric field components with different optical frequencies ($\omega_n$). As $\tau$ is scanned, $\tau$-varying interference between these electric field waves from the two pump pulses results in intensity modulations $I(\omega_n,\tau)$ with the corresponding frequencies. Consequently, the individual AuNRs with different aspect ratio ($R_n$), which are selectively excited with each $I(\omega_n,\tau)$, are tagged with the different temporal modulations with respect to $\tau$. The hot AuNRs created by subsequent relaxation processes such as electron heating during $T_w$ are then interrogated by the third (probe) pulse, $E_3(\omega')$. Then, a third-order nonlinear optical signal field, $E_s(\omega')$, with its wavevector parallel to that of the probe pulse is generated and spectral interference between the third-order 2DES signal field and probe pulse spectra, $E_s(\omega') + E_3(\omega')$, is recorded with a spectrometer combined with a charge coupled device (CCD) detector. Finally, the 2D electronic spectrum is then obtained by carrying out a Fourier transformation of the $\tau$-dependent interferogram $S(\tau,T_w,\omega')$ with respect to $\tau$, which provides the 2D spectrum, $S(\omega,T_w,\omega')$, in both $\omega$ (pump) and $\omega'$ (probe) frequencies[18,20–23,26].

To show that 2DES is capable of distinguishing between ensembles of AuNRs with different size (aspect ratio) distributions ($\Delta R$: standard deviation of $R$), we performed an illustrative simulation on 2DES in an ideal impulsive limit for three cases of AuNRs with large, intermediate, and small $\Delta R$'s and the results are depicted in Fig. 3. Assuming that the absorption spectrum can

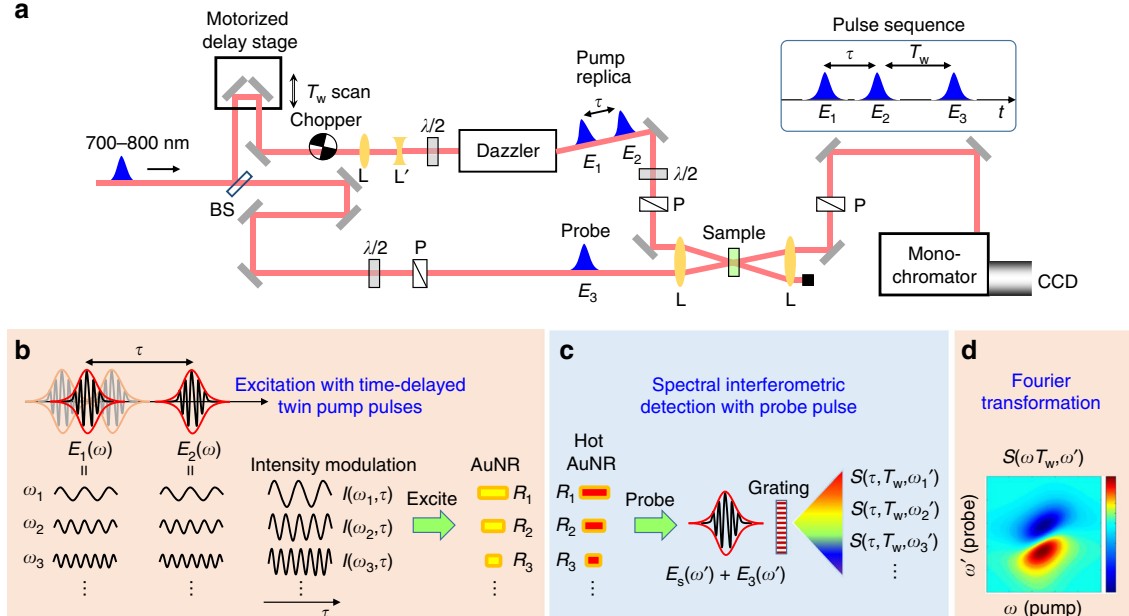

**Fig. 2** Experimental setup and basic concept of Fourier transform 2DES for AuNR. **a** Experimental scheme of our 2DES setup that utilizes a pulse shaper in two-beam geometry. BS beam splitter, L: convex lens, L′: concave lens, $\lambda/2$: half wave plate, P: polarizer, Dazzler: acousto-optic pulse-shaper (Fastlite), CCD: charge coupled device detector. A single femtosecond laser pulse is split into two: one is further duplicated as $E_1$ and $E_2$, which are used as time-delayed twin pump pulses, by the pulse shaper (Dazzler) and the other is used as a third (probe) pulse ($E_3$). The pulse sequence is $E_1$–$E_2$–$E_3$ in time order, where the time delay ($\tau$) between $E_1$ and $E_2$ is scanned by the pulse shaper and that ($T_w$) between $E_2$ and $E_3$ is controlled by a motorized delay stage. **b** Excitation with time-delayed twin pump pulses: As the time delay ($\tau$) between the twin pump pulses ($E_1(\omega)$ and $E_2(\omega)$) is scanned, $\tau$-varying interference of their constituting electric field waves with different optical frequencies ($\omega_n$) leads to intensity modulations, $I(\omega_n,\tau)$, of the pump beam with the same frequencies. Each $I(\omega_n,\tau)$ excites AuNRs with the aspect ratio ($R_n$) that can absorb it and tags its modulation on them. **c** Spectral interferometric detection with probe pulse: The hot AuNRs, which undergo spectral red-shift and broadening by photothermal relaxations after a waiting time ($T_w$) upon photoexcitation in **b**, is then probed by the third (probe) pulse ($E_3(\omega')$), which generates a third-order signal field ($E_s(\omega')$). The spectral interferometric signal, $S(\tau,T_w,\omega'_n)$, of $E_s(\omega') + E_3(\omega')$ is detected with a spectrometer. **d** Fourier transformation: The $\tau$-dependent interferogram, $S(\tau,T_w,\omega'_n)$, is Fourier transformed with respect to $\tau$ to demodulate the $\tau$-varying signals of the AuNRs tagged with the different pump modulation ($\omega_n$) during the excitation period and finally yield the 2DES spectrum, $S(\omega,T_w,\omega')$

be represented by a Voigt profile, resulting from the convolution of homogeneous (Lorentzian with $\Delta\omega_h$) and inhomogeneous (Gaussian with $\Delta\omega_{in}$) broadenings, one can define the degree of inhomogeneity as the ratio ($\beta$) of their line broadenings, that is, $\beta = \Delta\omega_{in}/\Delta\omega_h$ (Fig. 3a–c). After photoexcitation of the entire LgSPR band, the different subsets of AuNRs undergo electron heating, which gives rise to the red-shift and spectral broadening of the individual homogeneous SPR peaks. As a result, positive and negative absorption changes ($\Delta A$) at lower and higher probe-frequency ($y$-axis) regions, respectively, appear on the 2DES (Fig. 3d–f). Notably, it is shown that the NLS, which is the slope of the line connecting zero-crossing (nodal) points of $\Delta A$'s, is very sensitive to the degree of size inhomogeneity ($\beta$), decreasing from NLS = 0.87 to 0.11 as $\beta$ varies from $\beta = 3$ to 0.33. This indicates that the NLS of 2DES can be an excellent measure of the size inhomogeneity of AuNRs with different aspect ratios.

In 2DES, since the spectrally resolved pump beam can selectively photo-excite AuNRs of varying sizes, their individual pump-induced TA changes can be obtained from the probe spectra along the $y$-axis. Figure 3g plots the slice probe spectra taken at five different pump frequencies of the 2DES for the inhomogeneous case with $\beta = 3$ (Fig. 3d). The nodal points in Fig. 3g resulting from the individual TA spectra of different sized AuNRs are clearly separated in frequency and red-shifted with the pump frequency. In contrast, for the projected spectra of all three cases with different $\beta$ values onto the probe frequency axis, which are identical to the one-dimensional (in probe frequency) pump-probe TA spectra, their nodal point frequencies are almost

constant regardless of and thus insensitive to $\beta$ (Fig. 3h). This indicates that it is difficult to distinguish between AuNRs with different size inhomogeneous distributions by simply using the conventional pump-probe spectroscopy.

**2DES spectra and nodal line slope analysis.** The 2DES spectra (Fig. 4a–e) of the AuNRs were recorded as a function of $T_w$ up to 3 ps at different pump center wavelengths ($\lambda_{pumpc} = 700 \sim 820$ nm) covering the entire LgSPR band (see Supplementary Figure 1 and Movies 1–5, respectively, for all twelve 2DES spectra at $T_w = 1$ ps and for 2DES spectral evolution at various pump wavelengths). The five representative 2DES spectra at $T_w = 1$ ps (Fig. 4a–e) are superimposed onto a single 2D frequency space spanning the whole LgSPR band with a full-width-at-half-maximum (FWHM) of about 2100 cm$^{-1}$ (Fig. 4f). The positive 2DES peaks appearing below the diagonal line and the red-shifted negative peaks along the probe wavelength arise from both spectral broadening and the red-shift of the nonlinear difference spectrum from ultrafast electron heating (<200 fs) after the photoexcitation of the LgSPR mode. The cancellation of these positive and negative peak signals in the overlap region produces a nodal line in each 2DES spectrum.

As shown in the simulations (Fig. 3), the nodal line slope (NLS) of 2DES is a measure of the degree of size (aspect ratio) inhomogeneity of AuNRs at a given waiting time[19,24]. We obtained the NLS from a linear fit to the nodal points in the 2DES at the pump center wavelength and found that the NLS values at

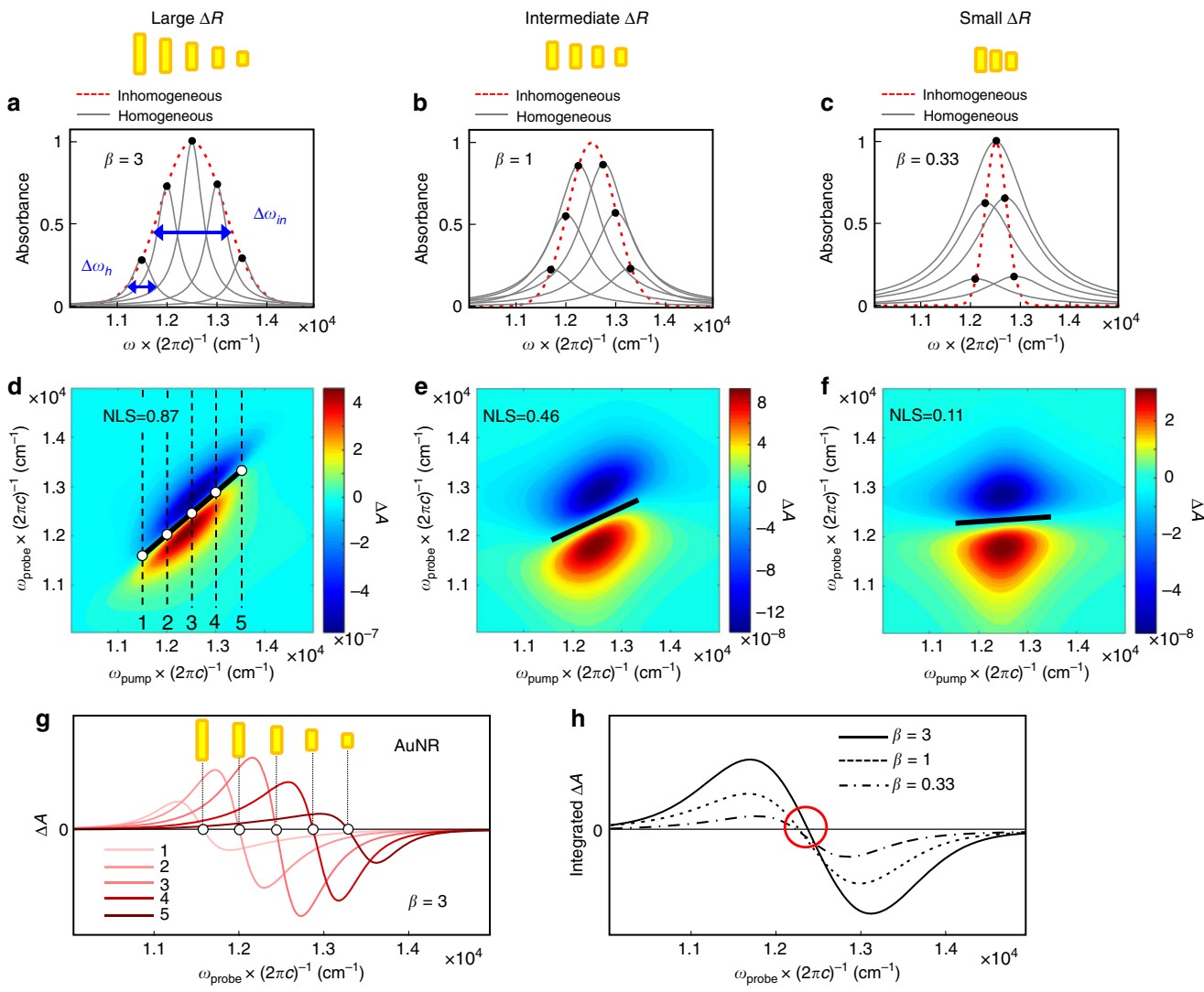

**Fig. 3** Simulation on 2DES spectra and nodal line slopes of AuNRs with different size inhomogeneity. **a–c** Contributions of homogeneous (gray solid lines) and inhomogeneous (red dashed lines) line broadenings to the LgSPR band of AuNRs with large (**a**), intermediate (**b**), and small (**c**) aspect ratio distributions ($\Delta R$: standard deviation of the aspect ratio ($R$) distribution). The homogeneous and inhomogeneous line shapes are assumed to be Lorentzian, $L_n(\omega) = L_0(\Delta\omega_h) \times (2\pi)^{-1} \times [(\omega - \omega_n)^2 + (\Delta\omega_h / 2)^2]^{-1}$, and Gaussian functions, $G(\omega) = G_0\exp[-4\ln(2) \times (\omega - \omega_0)^2 \times (\Delta\omega_{in})^{-2}]$, respectively, where $L_0$ and $G_0$ are the amplitudes and $\Delta\omega_h$ and $\Delta\omega_{in}$ are the FWHMs of $L(\omega)$ and $G(\omega)$, respectively. Here we define the degree of inhomogeneity ($\beta$) as the ratio of inhomogeneous ($\Delta\omega_{in}$) to homogeneous ($\Delta\omega_h$) broadenings, that is, $\beta = \Delta\omega_{in} / \Delta\omega_h$. **d–f** 2DES spectra calculated in an impulsive limit, where the laser pulse spectra are broad enough to cover the entire LgSPR band, for $\beta = 3$ (**d**), 1 (**e**), and 0.33 (**f**). It is assumed that each homogeneous peak after electron heating of AuNRs undergoes a red-shift of the peak by 30 cm$^{-1}$ and a spectral broadening by 2% of the original bandwidth ($\Delta\omega_h$). It is clearly shown that the nodal line slope (NLS: black solid line) is very sensitive to the degree of inhomogeneity and decreases with $\beta$. **g** The slice spectra (along the black dashed lines: 1–5 in **d**) taken at different pump frequencies ($\omega_{pump}$) of the 2DES spectrum for $\beta = 3$ (highly inhomogeneous case). These slice spectra and their separate nodal points (white circles) correspond to the ones produced by the spectral red-shift and broadening of individual AuNRs with different $R$. **h** The projected (integrated) spectra of the 2DES in **d–f** onto the probe axis (y-axis), which are essentially identical to the pump-probe TA spectra, for $\beta = 3$ (solid), 1 (dashed), 0.33 (dash-dotted). In contrast to the strong dependence of the NLS on $\beta$, the nodal points in the pump-probe TA spectra remain almost unchanged regardless of $\beta$ values (or degree of aspect ratio inhomogeneity) and cannot be of use to distinguish between different size inhomogeneity of AuNRs. Here, $c$ in the x–y labels represents the velocity of light

$T_w = 1$ ps strongly depends on the pump center wavelength (Fig. 4g). They reach a maximum (close to unity) around 780 nm, that is, the LgSPR band maximum, but decrease as the pump center wavelength ($\lambda_{pump_C}$) approaches to either the red or blue edges of the LgSPR band. This experimental observation suggests the following scenario on the inhomogeneous broadening of the LgSPR band of AuNRs. Since the frequency-tunable pump laser allows us to excite different subensembles of the heterogeneous AuNRs under investigation, the pump center wavelength-dependence of NLS shows the different degree and nature of

polydispersity of each subensemble selected by the relatively narrowband pump pulse (Fig. 4g). Here, a subensemble of the AuNRs excited by the pump with the finite bandwidth centered at the LgSPR band maximum (780 nm) has a more diverse and denser aspect ratio distribution of the AuNRs around the average $R \sim 3.84$, resulting in a more heterogeneous excitation of the AuNRs by the pump. On the contrary, the size distribution of the AuNRs is more discrete and the different homogeneous subensembles are less dense at both edges of the LgSPR band so that a less heterogeneous excitation of the AuNRs is induced

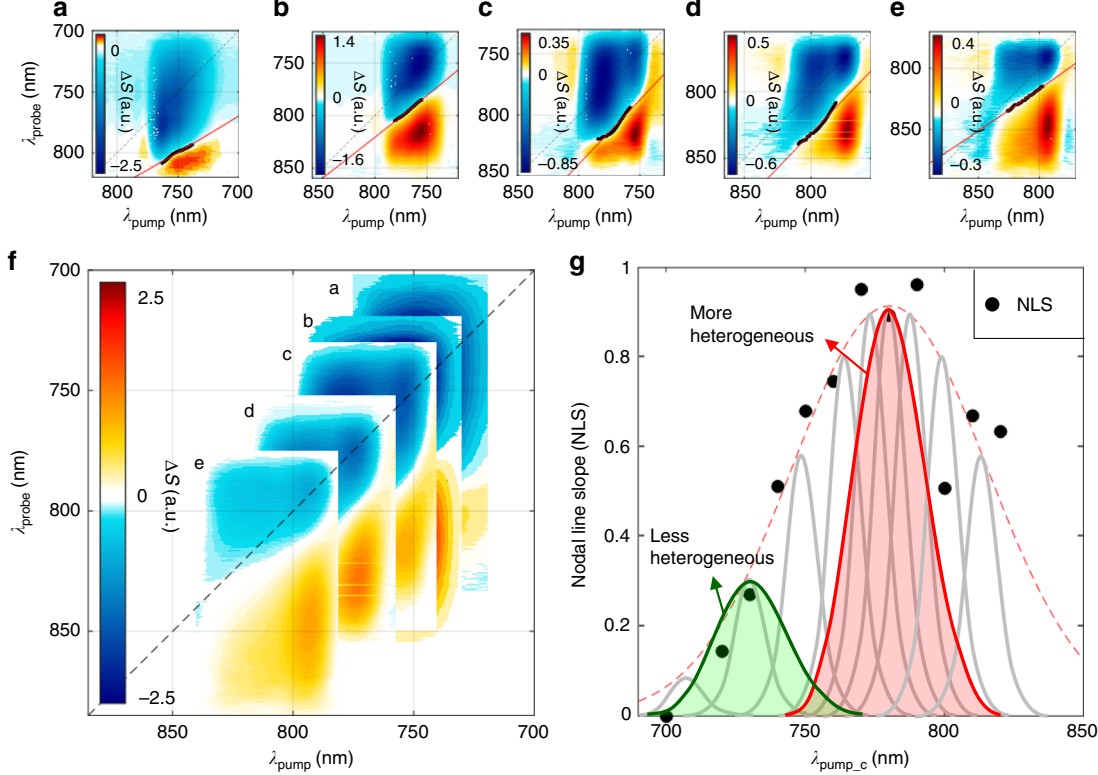

**Fig. 4** Experimental 2DES spectra and nodal line slopes at different pump center wavelengths. **a–e** Representative 2DES spectra of the AuNR measured at five different pump and probe center wavelengths. Note that the center wavelengths ($\lambda_{pump_c}$) of pump and probe spectra are the same and the 2DES signals plotted in all the figures (here and below) represents the differential transmitted spectra, $\Delta S(\omega_\tau, T_w, \omega_t)$ (see "Methods" section for more details). **a** 740 nm. **b** 760 nm. **c** 770 nm. **d** 780 nm. **e** 810 nm ($T_w = 1$ ps). The black and red solid lines are the experimentally measured nodal lines and their fits, respectively. **f** The five 2DES spectra of **a–e** superimposed onto a single 2D frequency space spanning the entire LgSPR band. The black dashed line represents the diagonal line. **g** The wavelength dependence of the nodal line slope (black circles) of the 2DES spectra of the AuNRs. The red dashed line represents a fit with a Gaussian at 780 nm with an FWHM of 1320 cm$^{-1}$, which approximately matches the LgSPR band (Fig. 1a). The inhomogeneous distribution of the individual homogeneous LgSPR peaks (gray solid lines) for AuNRs with different aspect ratios is schematically drawn along with the narrowband pump spectra tuned to the LgSPR band maximum (red shaded area) and to the blue edge (green-shaded area). A denser distribution of AuNRs with different aspect ratios around 780 nm leads to a more heterogeneous excitation of AuNRs by the pump, producing a high NLS. In contrast, at both edges of the LgSPR band, the inhomogeneous subensembles of AuNRs are spectrally separated and comparatively sparse, so the degree of heterogeneity of AuNRs that can be excited by the pump is lower than that at the LgSPR peak position (red shaded area), which results in a small NLS

by the pump tuned to that band edge. According to this scenario, it becomes understandable that the NLS value approaches to zero as the pump center wavelength is blue-shifted because the bandwidth (about 850 cm$^{-1}$) of the pump beam is close to typical intrinsic SPR linewidths of monodisperse AuNRs that are about 0.08 eV (640 cm$^{-1}$)[27] or 0.09 eV (720 cm$^{-1}$)[1,28] for gold nanorods with an aspect ratio of about 4, allowing an excitation of relatively homogeneously distributed AuNRs.

Another important result from the NLS analysis of our 2DES spectra is the time-dependent decay of NLS (Fig. 5f). Due to either the scattered or diffracted pump beams interfering with the 2DES signal and the probe beams at the detector or electronic coherent artifacts, a complicated spectral pattern appears in the Fourier-transformed 2DES spectra at very short waiting times, which is also found in short-time TA spectra (see Supplementary Figure 2 for the short-time TA spectra up to 0.5 ps). In addition, due to the finite duration of our slightly chirped pulse, it is somewhat difficult to identify the nodal line at waiting times <100 fs. Nonetheless, the transient behavior of NLS at <300 fs is attributable to ultrafast electron dynamics, and the NLS decays monotonically with a time constant of about 5 ps, which is the rate of homogeneous line-broadening or dephasing that results from e–ph couplings.

To further investigate the short-time behavior of a nodal line extracted from a chosen 2DES spectrum, the probe wavelength at the nodal point (nodal point wavelength) at which the nodal line crosses the vertical line at the center wavelength ($\lambda_{pump} = 760$ nm) of the pump pulse (Fig. 5a–e) is plotted with respect to waiting time in Fig. 5f. The initial decrease in the nodal point wavelength (NPW) at $T_w < 100$ fs results from the e–e scattering-induced dephasing of the initial coherent and collective oscillations of plasmonic electrons in each AuNR. As waiting time increases ($T_w > 100$ fs), the NPW undergoes a slow blue-shift caused by homogeneous relaxation due to e–ph couplings. Indeed, ultrafast electron heating is manifested by the time-evolution of 2DES at less than 300 fs (see Supplementary Movies 1–5 for 2DES spectral evolutions at different wavelengths). Because the NLS data from time-resolved 2DES spectra are free from inhomogeneous line-broadening contributions[19,24], the ultrafast transient behaviors of NLS and NPW are direct signatures of the electron heating resulting from e–e scattering process in sub-100 fs timescale induced by the absorption of pump photons. It is believed that this is the first experimental and direct observation of electron dynamics and photothermal relaxation with heterogeneity-free and phase-sensitive (not absolute) 2DES in combination with a tunable femtosecond laser.

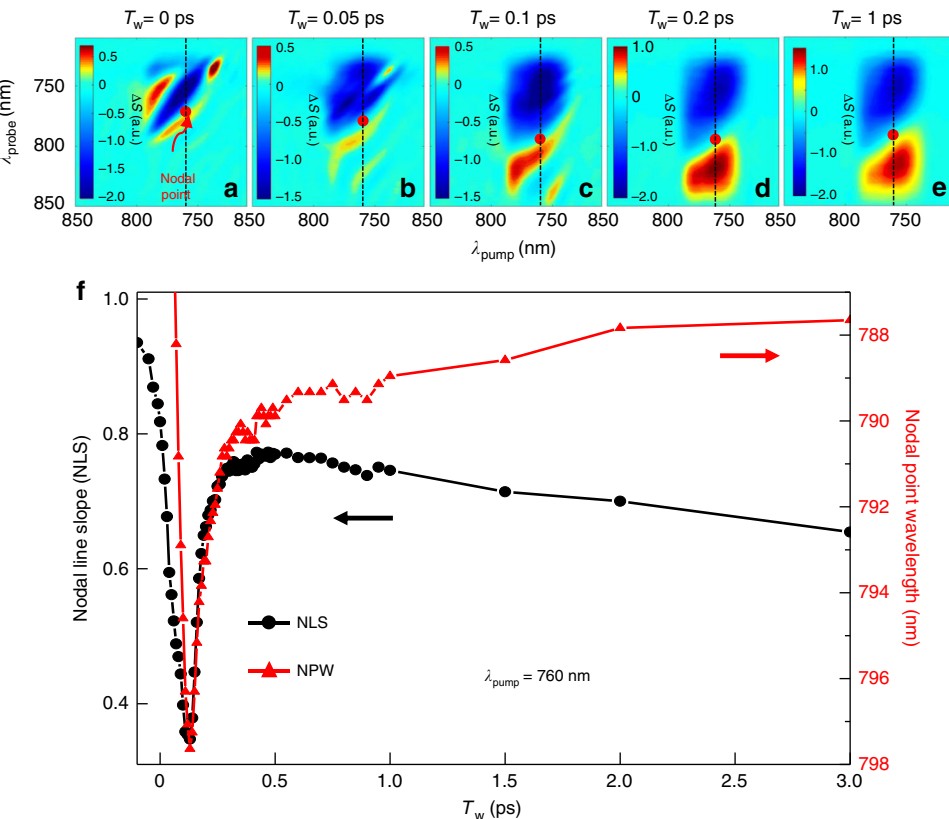

**Fig. 5** Time-dependent nodal line slope and nodal point wavelength. **a–e** Time-dependent 2DES spectra obtained with $\lambda_{pump_c} = 760$ nm of pump and probe pulses at waiting times of $T_w = 0$ ps (**a**), 0.05 ps (**b**), 0.1 ps (**c**), 0.2 ps (**d**) and 1 ps (**e**). **f** Time-dependent nodal line slope (NLS: black circles and line) values and nodal point wavelength (NPW: red triangles and line) extracted from the 2DES spectra from $T_w = -0.1$ ps to $T_w = 3$ ps, including **a–e**. The NPW taken at $\lambda_{pump} = 760$ nm (dashed line) in the 2DES spectra (**a–e**) is plotted as a function of the waiting time

**Spectral interference by transient grating in 2DES.** The 2DES spectrum displayed on excitation ($\omega_{\tau'}$) and emission ($\omega_t$) frequency axes can provide additional information on the spectral interference between the collinearly propagating pulses at the spectrometer (Fig. 6a–d). The two-dimensional fringe patterns become pronounced particularly at negative waiting times ($T_w < 0$), where the probe pulse (pr; $E_3$) arrives at the sample before the second pump pulse (pu2; $E_2$) among the twin pump pulses produced by a pulse shaper (Fig. 2a). The fringe spacing becomes smaller at larger $|T_w|$ (Fig. 6j and Supplementary Figure 5). This phenomenon can be explained by another, yet different four-wave mixing process called transient grating (Fig. 6l)[24,29]. As the time delay ($\tau$) of the first pump pulse (pu1; $E_1$) is scanned with respect to the second pump (pu2) by employing a pulse shaper, the pair of the overlapping pulses, pu1 and pr, that propagate non-collinearly in space, interacts with the AuNRs to create a temporal and spatial transient grating (TG) across the sample due to a spatially regular and interferometric excitation of the coherent SPR or electron heating (Supplementary Note 1). Then, the subsequent second pump (pu2) interacts with thus created TG and the resulting third-order TG signal field with wavevector of $\mathbf{k}_{TG}$ is then diffracted into the propagation direction ($\mathbf{k}_{pr}$) of the probe (pr) because the phase-matching condition is $\mathbf{k}_{TG} = \mathbf{k}_{pr}$ in either possible pulse sequence, pr-pu1-pu2 ($\mathbf{k}_{TG} = \mathbf{k}_{pr} - \mathbf{k}_{pu1} + \mathbf{k}_{pu2}$) or pu1-pr-pu2 ($\mathbf{k}_{TG} = -\mathbf{k}_{pu1} + \mathbf{k}_{pr} + \mathbf{k}_{pu2}$) during the $\tau$-scan at a fixed negative $T_w$. The delay time of the collinearly propagating TG signal field from the probe pulse is approximately $|T_w|$, and the two fields interfere with each other to produce a spectral interferogram with the fringe spacing of $1/|T_w|$ on the frequency-resolved CCD along $\omega_t$-axis (probe axis) (Supplementary Figures 6 and 7). Similar to Fig. 2b, at a negative waiting time,

scanning the time delay ($\tau' = |T_w| - |\tau|$) between pu1 and pr gives rise to temporal modulations of the spatial phase of the TG field induced by the interference of the excitation fields (pu1 and pr) with various optical frequencies ($\omega_{\tau'}$), which can individually excite the inhomogeneous subensembles of AuNRs with different aspect ratios. Then, a Fourier transformation of the spectral interferogram, $S(\tau', T_w, \omega_t)$, created by the modulated TG signal interfering with the preceding probe field, with respect to $\tau'$ yields the spectral interferometric 2DES spectra, $S(\omega_{\tau'}, T_w, \omega_t)$, shown in Fig. 6a–d.

It should be noted that despite employing perpendicular detection scheme with two polarizers having an extinction ratio of about $10^{-4}$ to $10^{-5}$ to remove any contamination of the 2DES signal from stray pump pulses scattered or TG signal diffracted by the AuNRs, very strong 2D spectral fringe patterns were still observed at negative waiting times (Fig. 6a–d). Due to the coherent SPR effect of AuNRs, the interfering pu1 and pr fields induce an anomalously strong nonlinear electronic response, which considerably modulates the refractive index of AuNRs in a regular spacing across the sample, giving rise to a strong diffraction of the TG signal field. The key for this extraordinarily large but transient diffraction signal of AuNRs is the free electron-like behaviors of SPR-excited electrons during the interaction time of both pu1 and pr. This is in stark contrast to organic molecules with tightly bound electrons to nuclei. Indeed, the 2DES of the bio-organic system measured with the same setup did not show any strong and dense 2D fringe patterns at similar negative waiting times (Supplementary Figures 3 and 4 and Supplementary Methods).

One of the noticeable features in the spectral interferometric 2DES spectra at $T_w < 0$ is that the fringe patterns are roughly

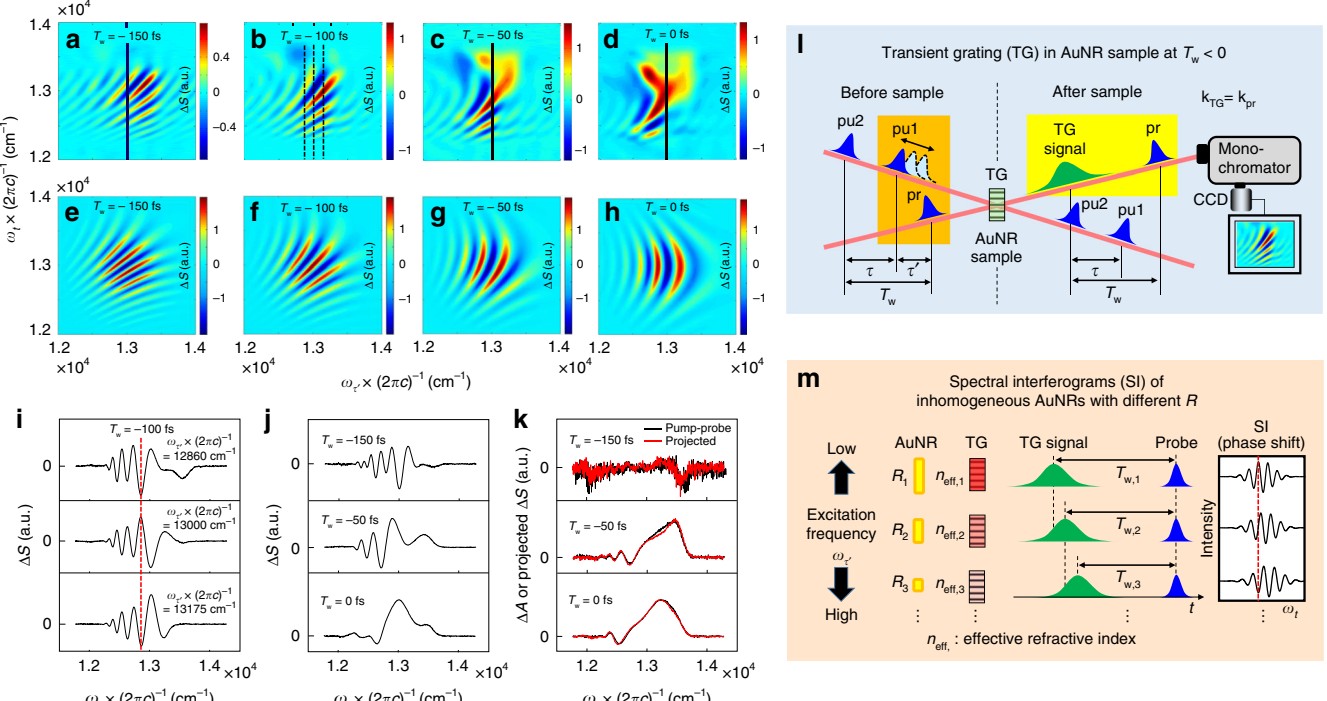

**Fig. 6** Spectral interference patterns by transient grating in 2DES spectra. **a–d** The experimental 2DES spectra ($\Delta S$) of the AuNRs measured at waiting times ($T_w$) of −150 (**a**), −100 (**b**), −50 (**c**), and 0 fs (**d**) are shown. **e–h** The simulated 2DES spectra based on another four-wave mixing scheme called transient grating (see the Methods for more details). **i** The slice interferometric spectra taken at three different excitation frequencies, $\omega_{\tau'} \times (2\pi c)^{-1} =$ 12860 cm$^{-1}$, 13000 cm$^{-1}$ and 13175 cm$^{-1}$ of the 2DES spectra at $T_w = -100$ fs (vertical dashed lines in **b**). **j** The slice spectra taken at the center frequencies (vertical solid lines in **a**, **c**, and **d**) in the 2DES spectra ($T_w = -150$, −50 and 0 fs). **k** The pump-probe TA ($\Delta A$: black lines) and projected ($\Delta S$: red lines) spectra onto the $\omega_t$-axis in the 2DES spectra at the corresponding waiting times ($T_w$) are plotted together for comparison. **l** Transient grating (TG) process in the AuNR sample at $T_w < 0$. At negative waiting times, the probe pulse (pr) arrives at the sample before the second pump pulse (pu2). The overlapping pu1 and pr (orange area) interfere with each other to create a TG in the nonlinear optical sample. The following pu2 is then diffracted by the TG into the propagation direction of the pr pulse ($\mathbf{k}_{TG} = \mathbf{k}_{pr}$). The TG signal field is then spectral-interferometrically detected at the spectrometer. **m** Phase-shifted spectral interferograms of inhomogeneous AuNRs with different aspect ratios ($R_j$). The TG-inducing beams (pu1 and pr) with various excitation frequencies ($\omega_{\tau'}$) individually excite the AuNRs with different $R_j$. The TG, which essentially represents the spatially modulated refractive index by the strong SPR effect of AuNRs, can further time-delay the TG signal by modulating its velocity. Due to the different size effect of the inhomogeneous AuNRs, they can have different effective refractive indices ($n_{\text{eff},j}$). The TG signal field created at lower excitation frequency undergoes the larger $n_{\text{eff}}$ and then is slightly more delayed ($T_{w,j}$) from the probe pulse than that at higher excitation frequency undergoing the smaller $n_{\text{eff}}$, resulting in the $\omega_{\tau'}$-dependent phase shift of the spectral interferograms

diagonally elongated (Fig. 6a), which means that there exists some phase shifts between the spectral interferograms (along $y$-axis) at different excitation frequencies ($\omega_{\tau'}$). Figure 6m explains how such phase shifts can arise in the 2DES fringe pattern of AuNRs. The TG, created by a spatial modulation of the refractive index, not only can diffract the pu2 beam in space, but also can further change the delay time ($T_w$) between the third-order (in electric field) TG signal field and the probe pulse by altering the velocity of the diffracted TG signal field at the sample. Since the subsets of AuNRs with different aspect ratios ($R_j$) can be individually excited by the TG-inducing beams with various excitation frequencies ($\omega_{\tau'}$), they can have different effective refractive indices ($n_{\text{eff,j}}$) upon photoexcitation, resulting in $\omega_{\tau'}$-dependent (inhomogeneous) delay times ($T_{w,j}$) and consequently in phase-shifted spectral interferograms. The simulated 2DES spectra in Fig. 6e–h are in excellent agreement with the experimentally measured 2D fringe spectra in Fig. 6a-d. Here, the simulated spectra in Fig. 6e-h were obtained from the calculated spectral interferograms of the TG signal field and the probe pulse with various $T_w$ values with respect to $\omega_{\tau'}$—note that the $\omega_{\tau'}$-dependence of $T_w$ resulting from the inhomogeneous distribution of $n_{\text{eff}}$ for AuNRs with different aspect ratios. We here assumed a quadratic increase of $T_w(\omega_{\tau'})$ as decreasing the excitation frequency.

Similar to the experimental results in Fig. 6a–d, they exhibit the diagonally elongated fringe patterns with unequal spacing (wider spacing at higher $\omega_{\tau'}$ along the diagonal line) in the 2DES spectra at $T_w = -150$ and −100 fs and the vertically elongated but parabolic-curved fringe patterns due to the group delay dispersion (GDD = 800 fs$^2$) of the pulse even at $T_w = 0$ fs. On closer inspection of the slice interferometric spectra taken at three different excitation frequencies, $\omega_{\tau'} \times (2\pi c)^{-1} = 12860$, 13000, and 13175 cm$^{-1}$, of the 2DES spectra at $T_w = -100$ fs (Fig. 6b), roughly one cycle (about 2.6 fs for 13000 cm$^{-1}$) of the phase shift in the spectral interferogram occurs between $\omega_{\tau'} \times (2\pi c)^{-1} = 12860$ and 13175 cm$^{-1}$ (Fig. 6i), where $c$ is the velocity of light. This demonstrates that such a small difference ($\Delta T_w \approx 2.6$ fs) between the time delays of the TG signal field created by the inhomogeneous distribution of the AuNRs can be distinguished by carefully examining the 2DES fringe pattern even in the excitation frequency range of $\Delta \omega_{\tau'} \times (2\pi c)^{-1} \sim 300$ cm$^{-1}$ much narrower than typical intrinsic bandwidth (about 700 cm$^{-1}$) of the homogeneous broadening.

Interestingly, this SPR TG-induced-fringe pattern resolved in the 2DES spectrum is averaged out and almost disappears in the conventional pump-probe spectra, which is equivalent to the projected average one-dimensional spectrum onto the $\omega_t$-axis of

the 2DES spectrum (Fig. 6k)[30]. As shown in the 2DES (Fig. 6a), the fringe on the 2DES spectrum is diagonally elongated due to the inhomogeneous distribution of AuNRs and its phase is thus alternately switching to produce an oscillating positive and negative signal pattern along the $\omega_{\tau'}$-axis (pump frequency). As a result, the individual spectral fringes (along $y$-axis) of the different sized subensembles of AuNRs excited at different pump frequencies ($\omega_\tau$) are superimposed to cancel out when projected average onto the $\omega_t$-axis, which accounts for why the fringe information is completely lost in the conventional pump-probe TA ($\Delta A$) spectra for AuNRs with an inhomogeneously broadened size distribution.

## Discussion

The present 2DES study with nodal line slope and spectral interference pattern analyses provide direct information on the degree of size inhomogeneity of inhomogeneously distributed AuNRs and their homogeneous electron dynamics, which cannot be easily extracted from one-dimensional (in frequency) steady-state or pump-probe spectroscopy signals. In particular, the spectral fringe exclusively observed in the 2DES spectrum is direct evidence of exceptionally strong and transient nonlinear electronic responses by excited AuNR LgSPR modes, which cannot be observed with conventional pump-probe techniques due to their limited spectral resolvability compared to 2DES. Such significant nonlinear electronic coherence exhibited by AuNRs when their SPR modes are excited by the presence of another pulsed field could be used in future scattering-based microscopy applications. We anticipate that the present experimental results are of use in understanding and designing metallic nanorods with improved electronic and optical properties that can be optimized for various applications.

## Methods

**Pulse shaper-based 2DES setup.** The experimental details of our pulse shaper-based 2DES setup illustrated in Fig. 2a have been described elsewhere[26]. In brief, a Yb:KGW-doped amplifier system (PHAROS, Light Conversion) centered at 1030 nm with a repetition rate of 500 Hz is used to pump a non-collinear optical parametric amplifier (NOPA, ORPHEUS-N, Light Conversion), generating a laser pulse with a tunable center wavelength ranging from 700 to 820 nm, with a FWHM of approximately 80 nm. Each laser pulse is split into two pulses with a beam splitter. The first, which is used as the pump, is sent into an acousto-optic pulse-shaper (Dazzler, Fastlite), which produces a duplicated pulse pair, with a FWHM of approximately 60 nm and controls the variable time delay $\tau$ (coherence time) as well as the phase shift between these replica pulses in a programmable manner. The second pulse from the beam splitter is used as the probe and it is focused together with the pump beam into the sample using a fused silica convex lens with a focal length of 10 cm so that the pump and probe beams spatially overlap at the sample. A motorized delay stage on the pump beam path adjusts the time delay $T_w$ (waiting time) between the second pump and the probe pulses. The probe spectra transmitted through the sample in a quartz cell with a thickness of 1 mm are measured with a spectrometer (SP 2300i, PIXIS) equipped with a CCD (100B, PIXIS) as $\tau$ is scanned. For shot-by-shot measurements, data acquisition using the CCD is synchronized with the laser repetition rate (500 Hz). Two interferometric spectra, $S(\tau, \phi_{12}=0, T_w, \omega_t)$ and $S(\tau, \phi_{12}=\pi, T_w, \omega_t)$, where $\phi_{12}$ is the relative phase shift between the replica pump pulses, are recorded for consecutive pump pulses every $\tau$. Then, their difference is calculated (phase cycling scheme for measuring the absorptive signal) and its Fourier-transformation yields the differential transmitted 2DES spectra, $\Delta S(\omega_\tau, T_w, \omega_t) = S(\omega_\tau, \phi_{12}=0, T_w, \omega_t) - S(\omega_\tau, \phi_{12}=\pi, T_w, \omega_t)$[31]. The time resolution of our TA and 2DES experimental setups, which is determined by the cross-correlation of the pump and probe, varies from 110 to 150 fs depending on the center wavelength of the laser beam. To effectively minimize any undesired contribution from the pump photons scattered by the AuNRs to the self-heterodyned 2DES signal, we deliberately controlled the linear polarization states of the pump and probe beams so that they were perpendicular to each other. This orthogonal polarization scheme did not affect our 2DES measurement results because the timescales of both the AuNR reorientational motions and Förster resonant excitation transfers between neighboring AuNRs are much longer than the 2DES experimental time window of 3 ps. Note that our TA anisotropy remains constant in time up to 50 ps.

**Preparation and absorption spectrum of AuNRs sample.** The silica-coated AuNRs used in the present work (747971, Sigma Aldrich) have an axial diameter of 9–11 nm and a length of 35–41 nm (aspect ratio ~3.84) and are dispersed in pure water. We concentrated the solution in a 1 mm quartz sample cell until the absorption spectra reached an optical density of 0.8 at 760 nm (Fig. 1a). The absorption spectrum shows two peaks at 512 and 760 nm, which correspond to the TrSPR and LgSPR modes of the AuNRs, respectively. It is important to note that our laser pulse spectrum is narrower than the AuNR LgSPR band so not all of the AuNRs covering the LgSPR band are excited (Fig. 1a). Consequently, only a sub-ensemble of AuNRs with a narrow distribution of aspect ratios are selected by the finite pump spectral bandwidth and their ultrafast thermalization and relaxation processes are monitored by our femtosecond pump-probe TA and 2DES measurements.

**Simulation of spectral interferometric 2DES spectra by TG.** In our numerical simulations of the TG-induced 2DES spectra shown in Fig. 6e–h, the 2DES spectra at given waiting times $T_w = -150$ (Fig. 6e), $-100$ (Fig. 6f), $-50$ (Fig. 6g), and 0 fs (Fig. 6h) were obtained from the spectral interferograms (along the probe axis) of time-separated two Gaussian electric fields, $E_{TG}$ (TG signal) and $E_{pr}$ (probe), for each excitation frequency, $\omega_{\tau'j}$ (see Fig. 6m). The TG signal is assumed to be a transform-limited pulse whose Fourier-transformed spectrum has a center frequency of $\omega_0 \times (2\pi c)^{-1} = 12987$ cm$^{-1}$ and a FWHM of $\Delta\omega_{TG} \times (2\pi c)^{-1} = 850$ cm$^{-1}$, whereas the probe pulse to be slightly chirped with a FWHM of $\Delta\omega_{pr} \times (2\pi c)^{-1} = 850$ cm$^{-1}$ and a second-order dispersion of GDD = 800 fs$^2$. In the frequency domain, therefore, the probe and time-delayed TG signal field spectra are given respectively as $E_{pr}(\omega_t) = E_{pr,0} \times \exp[-2\ln(2) \times (\omega_t - \omega_0)^2 \times (\Delta\omega_{pr})^{-2}] \times \exp[i \times \text{GDD} \times (\omega_t - \omega_0)^2 / 2]$ and $E_{TG}(\omega_t, T_{wj}) = E_{TG,0} \times \exp[-2\ln(2) \times (\omega_t - \omega_0)^2 \times (\Delta\omega_{TG})^{-2}] \times \exp(i\omega_t T_{wj})$, where $i$ is the imaginary number and $E_{pr,0}$ and $E_{TG,0}$ are the complex amplitudes of the probe and TG signal fields, respectively. $T_{w,j}$ is the time delay between $E_{pr}$ and $E_{TG}$ for the j-th excitation frequency component ($\omega_{\tau'j}$) and given as $T_{w,j} = T_w + \Delta T_j$, where $\Delta T_j = 0.08$ fs $\times (j - 201)^2 / 200$ for $\omega_{\tau'j} \times (2\pi c)^{-1} = 12000$ cm$^{-1} + (j-1) \times 10$ cm$^{-1}$ ranging from 12000 cm$^{-1}$ (j = 1) to 13990 cm$^{-1}$ (j = 200). The quadratic dependence of $\Delta T_j$ with $\omega_{\tau'}$ was used for a better fit to the experimental data in Fig. 6a–d. The difference between two spectral interferograms along the probe axis ($\omega_t$), $S(\omega_{\tau'j}, T_w, \omega_t, \phi_{12}=0) = |E_{pr}(\omega_t) + E_{TG}(\omega_t, T_{wj})|^2$ and $S(\omega_{\tau'j}, T_w, \omega_t, \phi_{12}=\pi) = |E_{pr}(\omega_t) - E_{TG}(\omega_t, T_{wj})|^2$, for each $\omega_{\tau'j}$ is calculated to construct a differential 2DES spectrum, $\Delta S(\omega_{\tau'j}, T_w, \omega_t) = S(\omega_{\tau'j}, T_w, \omega_t, \phi_{12}=0) - S(\omega_{\tau'j}, T_w, \omega_t, \phi_{12}=\pi)$.

**Data availability.** All relevant data are available from the corresponding authors (hjrhee@kbsi.re.kr and mcho@korea.ac.kr) upon request.

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

## Acknowledgements

This work was supported by IBS-R023-D1. H.R. thanks financial support from IBS CMSD (PB2016005) and KBSI (E37300). The 2DES experimental data presented here were measured using the Femtosecond Multi-dimensional Laser Spectroscopic System (FMLS) at the Korea Basic Science Institute (KBSI). We thank Heejeong Kim, Dr. Jongbok Seo, and Dr. Chiyong Eom for providing LHC chromatophores of photosynthetic bacteria whose 2DES spectra are used to compare with those of AuNRs in the Supplementary Information.

## Author contributions

M.C. designed the overall investigation. A.L. and C.-S.H. carried out the experiments and contributed equally to this work. A.L., H.R and M.C. interpreted the experimental results and wrote the manuscript.
