## [Peer Review File · Nature Communications]

Reviewers' comments:

Reviewer #1 (Remarks to the Author):

This paper reports a one- and two-dimensional spectroscopic study of surface plasmon dynamics in gold nanorods. The experiments seem carefully done and the insight from the nodal line slope analysis is useful. However, the paper has too little discussion of the physics behind the main results and too much information on results of lower interest to make it acceptable in Nature Communications without major revision. In its current form it would be suitable for a more specialized journal. Further comments:

1. The authors note that sample heterogeneity rules interpretation of transient absorption data difficult, but are silent on how the same heterogeneity influences the times in Table 1. Do the authors regard these times as physically meaningful?
2. The optical Kerr effect is mentioned only in the caption of Figure 4. The main text should make it clear that their experiment set up makes their signal sensitive to this effect.
3. It is not always clear whether the authors are referring to the laser center wavelength or the excitation wavelength coordinate in their 2D spectra.
4. The extensive discussion of phenomena observed for $T < 100\text{fs}$ does not appear warranted given that the laser pulses have a duration of 150fs . Pulse overlap and scattering effects are well known.
5. There is a large volume of irrelevant supplementary information.

Reviewer #3 (Remarks to the Author):

The authors present 2D four wave mixing (2D-FWM) spectroscopy of an ensemble of gold nanorods. Their method sheds light on the homogeneous linewidth of nanorods in a broader, inhomogeneous distribution. The use of 2D spectroscopy in combination with plasmonic nanoparticles is in my opinion new and original. However, the results are poorly presented and remain qualitative, with lack of a good introduction to an audience who might not all be experts in 2D nonlinear techniques. It is not clear what is the particular main result which allows them to draw more information from the 2D-FWM experiment as their method follows an intuition drawn from molecular 2D experiments which is not easily extrapolated to the system under study without a good explanation. Therefore, while I find the topic of interest, I do not recommend publication of the work in its current form.

1. The manuscript does not explain in a self-contained and clear way what is the motivation of this particular experimental design for studying gold nanorods. It appears they have taken a setup used for molecular 2D-FWM and put in the gold nanorods. In conventional 2D spectroscopy in molecular systems there is a clear advantage as the off-diagonal elements say something about the coupling between different parts of the system. In the current situation, there are no coupled parts of the system and therefore there is no clear off-diagonal signal. Moreover, the role of the two pump pulses separated by a time τ is not very clear in this case, I understand this probes the FWM efficiency versus delay time but what is the meaning of the FT of this signal in the case of a gold nanorod subjected to a time-dependent hot-electron perturbation? The authors leave pretty much in the dark here what is the physical meaning of the signal and assume that all readers are experts in 2D fourier nonlinear spectroscopy. The reviewer for one is not an expert in this technique and has a very hard time following the line of reasoning presented by the authors. I think they need to do some more work to make the paper more self-explanatory if they want to interest an audience from e.g. the plasmonics community where most readers will need more information which is not scattered in many different specialist papers.

2. The role of the nodal slope is badly explained in the paper, as they just refer to previous works

where "the nodal line slope (NLS) is known to be a good measure of spectral inhomogeneity in transition frequencies for a given waiting time". I think this kind of effect has to be explained much better in the context of the system under study as it seems to be the central claim of this paper. To the reviewer it is not clear that the zero crossing and this slope is related to the homogeneous linewidth as it is well known that for gold nanorods the ultrafast dynamics has components in both real and imaginary parts of the permittivity, which both depend on wavelength and delay time, see for example Phys Rev Lett. 2011, 107(5):057402 and ACS Photonics, 2015, 2 (4), pp 521–529. Thus, the position of the zero and its slope will contain components related to the time dependent dispersion of the material nonlinearity.

3. The mixing of the scattered pump and degenerate probe appears interesting with rich effects which seem to have particular oscillation patterns. However these effects are not quantified. It is not clear if these are just spurious backgrounds which basically render the technique useless for short times, or if they contain any information, e.g. on fast time dynamics.

Some minor observations:

4. They should include clearly what is the observable plotted. Currently all the 2D color maps and the captions are missing the information about what are the units and what is the observable.

5. On Page 9 they state: "Due to the coherent SPR effect of AuNRs, the preceding probe pulse induces an anomalously strong nonlinear electronic response, which considerably modulates the refractive index of AuNRs." For the reviewer this paragraph is not clear, they should make another effort at explaining what is the effect that is observed here. How does the coherent SPR effect induce a strong nonlinear response? I believe this is the probe so how can this be the case? What do they mean with anomalous? Which nonlinear electronic response is meant, an ultrafast coherent plasmonic polarisation within the plasmon dephasing time, or induced hot electron gas?

6. A Conclusion paragraph is missing

7. In general there is no effort at quantitative modelling of effects. Everything is a bit qualitative. Given the literature suggested in point 2., I would expect that they could provide some semi-quantitative description of the effect?

Point-by-point response to the reviewer comments:

Reviewer #1

Remarks to the Author:

This paper reports a one- and two-dimensional spectroscopic study of surface plasmon dynamics in gold nanorods. The experiments seem carefully done and the insight from the nodal line slope analysis is useful. However, the paper has too little discussion of the physics behind the main results and too much information on results of lower interest to make it acceptable in Nature Communications without major revision. In its current form it would be suitable for a more specialized journal. Further comments:

Response: According to the reviewer's comment, considerable changes have been made in the revised manuscript. As the reviewer correctly pointed out, the main results and new interesting findings of this work are that the nodal line slope (NLS) analysis and spectral interference patterns observed in the 2DES spectra can provide direct information not only on the degree of size heterogeneity but also on the homogeneous electron dynamics of gold nanorods, which cannot be observed by conventional pump-probe transient absorption technique. In the revised manuscript, we thus have shortened the results-and-discussion part of pump-probe spectroscopy section and put more emphasis and discussions on the physical origins, meanings, and interpretations of the NLS (as well as nodal point wavelength) and spectral interference found in our femtosecond 2DES spectra. In particular, we have added new simulation results, discussions and conceptual schemes (see new Figures 2, 3, and 6) explaining our new observations and results.

1. The authors note that sample heterogeneity rules interpretation of transient absorption data difficult, but are silent on how the same heterogeneity influences the times in Table 1. Do the authors regard these times as physically meaningful?

Response: Table 1 in the original manuscript summarizes the photothermal relaxation times of SPR-excited (SPR: surface plasmonic resonance) electrons in gold nanorods that were obtained by analyzing our transient absorption data. They are in good agreement with the results of other previous works but do not provide further new information by themselves except the well-known time scales of photothermal relaxation processes of gold nanorods. Therefore, we have moved the table to Supplementary Information (Table S1). However, it is noteworthy that all the relaxation times, oscillation periods and damping times do not significantly depend on the pump-probe center wavelength ($t_{e-ph}=2-5$ ps for electron-phonon relaxation, $t_{ph-ph}=50-80$ ps for phonon-phonon relaxation, $T_{osc}=50-60$ ps for oscillation period, and $t_{damp}=40-60$ ps for damping time of oscillation) over the entire SPR band (720~800 nm), indicating that they are rather insensitive to the size heterogeneity of the gold nanorods with relatively low polydispersity (axial diameter: 9-11 nm, length: 35-41 nm) in the present case. In fact, such excitation wavelength-independences of e-ph and ph-ph relaxations times of gold nanorods, which were obtained from pump-probe measurements, were already reported and confirmed before. However, they are in contrast to the observables (NLS and spectral interference) of 2DES that are highly sensitive to the sample heterogeneity even within the spectral bandwidth (~ 800 cm^{-1}) of the pump much narrower than that (>2000 cm^{-1}) of the SPR band. This clearly shows the advantage and sensitivity of 2DES technique capable of revealing small and finite sample heterogeneity. We have discussed this weakness (insensitivity) of the pump-probe TA spectroscopy section (pages 5-6) of the revised manuscript and added discussions on such exceptional sensitivity of the 2DES signal to the sample heterogeneity (new Figure 2).

2. The optical Kerr effect is mentioned only in the caption of Figure 4. The main text should make it clear that their experiment set up makes their signal sensitive to this effect.

Response: In the previous manuscript, we provided qualitative discussion about the spectral interference patterns observed in our 2DES spectra at negative waiting times ($T_w < 0$) and we alluded that they could result from the scattered pump field due to the strong optical Kerr effect of SPR-excited electrons in AuNRs. In the revised manuscript, we have presented quantitative and extensive discussions on the origin of the spectral interference in 2DES, which is essentially related to a four-wave mixing process, called *transient grating* (Refs. 21 and 30). At $T_w < 0$, the probe pulse (pr) arrives at the sample before the second pump pulse ($pu2$). Please note that, by definition, the waiting time is the difference in arrival times of the $pu2$ and pr pulsed fields. As the time delay (τ) of the first pump pulse ($pu1$) is scanned with respect to the second pump ($pu2$), the non-collinearly propagating $pu1$ and pr in space interacts with the AuNRs to create a transient grating (TG) across the sample due to a spatially regular and interferometric excitation of the coherent SPR or electron heating (new Figure 6f and 6g and new Supplementary note (pages 8-9) in revised Supplementary Information). Then, the following $pu2$ pulse goes through (interacts with) thus created TG and the resulting third-order TG signal (KTG) is then generated with its propagation direction (\mathbf{k}_{pr}) to be collinear with the probe (pr) pulse according to the phase-matching condition of $\mathbf{k}_{TG} = \mathbf{k}_{pr}$ for either possible pulse sequence, pr - $pu1$ - $pu2$ ($\mathbf{k}_{TG} = \mathbf{k}_{pr} - \mathbf{k}_{pu1} + \mathbf{k}_{pu2}$) or $pu1$ - pr - $pu2$ ($\mathbf{k}_{TG} = -\mathbf{k}_{pu1} + \mathbf{k}_{pr} + \mathbf{k}_{pu2}$) (please see Figure 6f and Supp. Info.). This collinearly propagating but time-separated TG signal and probe by $|T_w|$ interfere with each other to produce a spectral interferogram where the fringe spacing is $1/|T_w|$. Essentially, since the generation of TG can be attributed to the modulation of the refractive index of interacting gold nanorods, it is considered that the observed interferometric signal in our setup is sensitive to the optical Kerr effect due to exceptionally strong plasmonic polarization or electron heating process by hot electrons of gold nanorods, which can change the refractive index of the gold nanorod sample. To make this clear, we have added schemes of the TG generation and the corresponding observable in our setup as well as 2D simulation results (new Figure 6b) explaining the underlying physical processes that are directly compared with our experimental 2DES data (Figure 6a) in the revised manuscript (see the new subsection entitled “**Spectral interference by transient grating in 2DES**” in pages 10-13).

3. It is not always clear whether the authors are referring to the laser center wavelength or the excitation wavelength coordinate in their 2D spectra.

Response: We thank the reviewer for pointing out any possible confusion from not clearly differentiating between the pump center wavelength and the excitation wavelength coordinate in the 2DES spectra. We have clarified whether it means the pump center wavelength (λ_{pump_c}) or the excitation wavelength coordinate (λ_{pump}) in the 2DES spectra in the revised manuscript.

4. The extensive discussion of phenomena observed for $T < 100$ fs does not appear warranted given that the laser pulses have a duration of 150fs. Pulse overlap and scattering effects are well known.

Response: We partly agree with this comment. Our measurement with a pulse duration of 110-150 fs does not guarantee any quantitatively accurate interpretation or assignment of *faster* components than the pulse duration. However, this does not mean that interpretations of the observed signal at negative waiting times ($T_w < 0$) are useless or incorrect even though our limited pulse duration makes us difficult to clearly resolve ultrafast events faster than ~ 100 fs. As we tried to address the second issue (comment) raised by the reviewer 1, the observable at $T_w < 0$ is indeed related to the TG signal, which can be rather enhanced when pulses with finite durations significantly overlap with each other

in time. We have provided extensive discussion on the physical meaning of the TG-induced signal in connection with the inhomogeneity of gold nanorods in the main text of the revised manuscript (pages in 10-13) and added schemes (Schemes S1 and S2). A further discussion to explain what is different between the observed signals at positive (normal 2DES measurement) and negative (spectral interference by TG in 2DES) waiting times is newly provided in the revised Supplementary Information (SI) in pages 8 and 9.

5. There is a large volume of irrelevant supplementary information.

Response: We have transferred portions of Supplementary Information (2DES experimental setup and pump-probe TA spectra) to the main text and reduced the number of Supplementary figures. However, we really believe that the rest of Supplementary Information is still relevant to the results and discussions in the main text and would be of help for researchers who want to gain further insight and related information. We have added conceptual schemes (new Schemes S1 and S2 in SI) and a new note to the revised SI to provide discussion about why such spectral interference patterns in 2DES spectra are pronounced at negative waiting times ($T_w < 0$) as well as why it is different from normal 2DES measurement at positive waiting times ($T_w > 0$). We believe that this should be useful in better understanding 2D fringe patterns observed here.

Reviewer #2

Remarks to the Author:

The authors present 2D four wave mixing (2D-FMW) spectroscopy of an ensemble of gold nanorods. Their method sheds light on the homogeneous linewidth of nanorods in a broader, inhomogeneous distribution. The use of 2D spectroscopy in combination with plasmonic nanoparticles is in my opinion new and original. However, the results are poorly presented and remain qualitative, with lack of a good introduction to an audience who might not all be experts in 2D nonlinear techniques. It is not clear what is the particular main result which allows them to draw more information from the 2D-FWM experiment as their method follows an intuition drawn from molecular 2D experiments which is not easily extrapolated to the system under study without a good explanation. Therefore, while I find the topic of interest, I do not recommend publication of the work in its current form.

Response: As we answered to the general comment of the reviewer 1, the main result of this work is that the nodal line slope (NLS) analysis results and spectral interference patterns found in our 2DES spectra can provide direct information on the degree of size inhomogeneity for an ensemble of gold nanorods as well as on their homogeneous electron dynamics, which cannot be extracted from one-dimensional (in frequency) steady-state or pump-probe spectroscopic signals. As the reviewer pointed out, however, they were qualitatively described and their physical meanings were not clearly presented in the previous (originally submitted) manuscript. Thus, we have newly added an introductory section of “**Two-dimensional electronic spectroscopy of AuNR**” (pages 6-8), which explains the physical meaning of the nodal points observed in 2DES and how the NLS is related to the degree of inhomogeneity of an ensemble of gold nanorods with various aspect ratios by presenting illustrative simulation results of their 2DES spectra (new Figure 2) in the revised manuscript. Furthermore, for those who are not familiar with the 2DES technique, we have also added the pulse shaper-based 2DES setup and presented a conceptual description on how the Fourier transform 2DES technique works with gold nanorods (our particular case) and on how homogeneous contributions of the individual subsets of gold nanorods with different aspect ratios (R_n) can be resolved on a 2D spectrum (new Figure 3). We hope that this newly added Introduction to 2DES and Figure 3 can be of some help for general readers who might not be experts in 2D nonlinear spectroscopy to better understand the basic concept of 2DES for gold nanorods (different from organic molecules with discrete electronic states) and the physical meaning of our experimental results.

1. The manuscript does not explain in a self-contained and clear way what is the motivation of this particular experimental design for studying gold nanorods. It appears they have taken a setup used for molecular 2D-FWM and put in the gold nanorods. In conventional 2D spectroscopy in molecular systems there is a clear advantage as the off-diagonal elements say something about the coupling between different parts of the system. In the current situation, there are no coupled parts of the system and therefore there is no clear off-diagonal signal. Moreover, the role of the two pump pulses separated by a time τ is not very clear in this case, I understand this probes the FWM efficiency versus delay time but what is the meaning of the FT of this signal in the case of a gold nanorod subjected to a time-dependent hot-electron perturbation? The authors leave pretty much in the dark here what is the physical meaning of the signal and assume that all readers are experts in 2D Fourier nonlinear spectroscopy. The reviewer for one is not an expert in this technique and has a very hard time following the line of reasoning presented by the authors. I think they need to do some more work to make the paper more self-explanatory if they want to interest an audience from e.g. the plasmonics community where most readers will need more information which is not scattered in many different specialist papers.

Response: In molecular systems like photosynthetic light-harvesting complexes, 2DES has been mainly used to investigate the electronic coupling and dynamics of such coupled multi-chromophore systems. As the reviewer correctly pointed out, it is well-known that off-diagonal peaks in 2DES spectra are particularly useful to characterize couplings, excitation transfers, molecular conformational changes, and so on, between the involved electronic states of a molecule or molecular complexes of interest. However, nodal line slope (NLS) is another well-known and useful observable extractable from 2DES spectra, which has also been proven to be of critical use in gaining information on the degree of dynamic inhomogeneity of solvated molecular systems (Refs. 16 and 21). Our motivation was to demonstrate that such NLS analysis can be applied to characterization of the size inhomogeneity of gold nanorods which clearly exhibit the nodal line between separate positive and negative peaks in 2DES spectra induced by hot electrons upon photoexcitation (please see the new Figure 2 with associated (newly added) description in the main text of the revised manuscript). A further explanation on the physical meaning of the NLS and the relation between the NLS and degree of inhomogeneity of gold nanorods is presented in our Response to the second comment of the reviewer below.

To make the paper comprehensible to readers from metallic nanoparticle and plasmonics communities, as the reviewer suggested, we have added a conceptual description of FT 2D spectroscopy method with the experimental layout of our pulse shaper-based 2DES setup in the revised manuscript (new Figure 3 and the text in pages 7-8). Basically, 2DES is to plot the third-order non-linear signals in two (independently controllable) frequency axes, excitation (ω_τ) and emission (ω_t) axes. Then, such two-dimensional spectra are recorded with respect to varying waiting times. In general, there are two ways to plot a 2D spectrum in the frequency and time domains: one (frequency domain) is to measure broadband probe (emission) spectra as varying the frequency (ω_τ) of narrowband excitation beams and the other (time domain) to measure broadband probe spectra as scanning the time delay (τ) between two broadband excitation pulses (twin pulses generated by the pulse shaper in our case). Current 2D spectroscopy experiments usually employ the time domain FT 2D method because the frequency domain method inevitably loses the time resolution. In our FT 2D method, the role of scanning τ is essentially to give the same effect as varying the excitation frequency (ω_τ) as shown in the new Figure 3b of the revised manuscript – please note the conjugate relationship between time and frequency via Fourier transformation. Each broadband excitation pulse, $E_1(\omega_\tau)$ and $E_2(\omega_\tau)$, can be decomposed into constituting electric field components with different optical frequencies (ω_n). As τ is scanned, τ -varying interference between these electric field waves from the two excitation pulses results in intensity modulations $I(\omega_n, \tau)$ with the corresponding frequencies. Consequently, the individual AuNRs with different aspect ratio (R_n) are selectively excited with each $I(\omega_n, \tau)$. That is to say, they are individually tagged with the intensity modulations with different temporal periods with respect to τ . After waiting time T_w , the excited hot electrons of AuNRs are then probed by the third (probe) pulse, $E_3(\omega_t)$. Finally, a FT of the τ -dependent interferogram $S(\tau, T_w, \omega_t)$ with respect to τ provides a 2D correlation spectrum, $S(\omega_\tau, T_w, \omega_t)$, between ω_τ (excitation or pump) and ω_t (emission or probe) frequencies.

In summary, the role of the τ -scan-&-FT of the signal in our method is to provide a way (1) to spectrally resolve broadband pump beams, then (2) to individually excite the gold nanorods with different aspect ratios, and finally (3) to spread the correlated emission spectra (ω_t) with each excitation frequency (ω_τ) on the 2D frequency axes. We have described this basic concept of the FT 2D method in more detail and provided detailed explanation on what the corresponding observable is, in the revised manuscript (see Figure 3 and the main text (pages 7-8)). We hope that the reviewer finds that newly added material and description are helpful for general readers who may not be familiar with 2DES of gold nanorods with sample heterogeneity.

2. The role of the nodal slope is badly explained in the paper, as they just refer to previous works where "the nodal line slope (NLS) is known to be a good measure of spectral inhomogeneity in transition frequencies for a given waiting time". I think this kind of effect has to be explained much better in the context of the system under study as it seems to be the central claim of this paper. To the reviewer it is not clear that the zero crossing and this slope is related to the homogeneous linewidth as it is well known that for gold nanorods the ultrafast dynamics has components in both real and imaginary parts of the permittivity, which both depend on wavelength and delay time, see for example *Phys Rev Lett.* 2011, 107(5):057402 and *ACS Photonics*, 2015, 2 (4), pp 521–529. Thus, the position of the zero and its slope will contain components related to the time dependent dispersion of the material nonlinearity.

Response: We agree with the reviewer that the physical meaning and significance of NLS were not clearly explained in the original manuscript. To clearly show the importance of the nodal line slope and how it is related to the degree of inhomogeneity ($\beta = \Delta\omega_n / \Delta\omega_h$, $\Delta\omega_n$: inhomogeneous linewidth, $\Delta\omega_h$: homogeneous linewidth) of gold nanorods, we have newly performed a 2D simulation for three representative cases of gold nanorod systems with large, intermediate, and small inhomogeneous size distributions and the simulation results for the sake of illustration have been discussed in the revised manuscript (new Figure 2 and the main text in pages 6-7). The simulation shows in a quantitative way that the NLS is very sensitive to the degree of size inhomogeneity (β) and its value decreases with β . This indicates that the NLS of 2DES can be a good measure of the size inhomogeneity of AuNRs with different aspect ratios. Furthermore, the spectrally resolved excitation beam can selectively photo-excite gold nanorods of varying sizes and their individual photothermal processes can be resolved in 2DES as explained in Figure 3b. The slice spectra along the y(probe)-axis at different pump frequencies thus represents the individual TA (transient absorption) spectra of the different sized gold nanorods (Figure 2c). This means that the transient behaviors of the nodal points which are separated along the pump frequency axis provide direct information on the homogeneous dynamics of excited electrons for each individual subensemble of gold nanorods with various aspect ratios. This is in stark contrast to the projected 2D (onto the y- or probe frequency axis) or conventional pump-probe spectra that average out the individual homogeneous contributions and thus cannot easily distinguish between the homogeneous and inhomogeneous cases (Figure 2d) – note that the nodal points from the conventional pump-probe spectra are not sensitive to the degree of inhomogeneity β , whereas the slope of nodal line from the 2DES spectra are to the degree of inhomogeneity. We have added such discussion in the revised manuscript. Also, the two papers mentioned by the reviewer (Baida, H. et al. *Phys. Rev. Lett.* **107**, 057402 (2011) and Zavelani-Rossi, M. et al. *ACS Photonics* **2**, 521-529 (2015)) are newly cited in the main text as references 24 and 25.

3. The mixing of the scattered pump and degenerate probe appears interesting with rich effects which seem to have particular oscillation patterns. However these effects are not quantified. It is not clear if these are just spurious backgrounds which basically render the technique useless for short times, or if they contain any information, e.g. on fast time dynamics.

Response: We thank the reviewer for pointing this out. The spectral interference (oscillation or fringe) patterns particularly at negative waiting times (when probe pulse arrives at the sample earlier by $|T_w|$ than the second pump pulse (pu2) among the twin pump pulses) are not spurious backgrounds but signatures attributed to another coherent four-wave mixing process called transient grating (TG). This conclusion is based on the following evidence. First, these oscillation patterns are not random at all and their fringe spacing along the probe axis (y-axis) decreases with $|T_w|$ (Figure 6d). This is fully consistent with the scenario of spectral interference between the time-separated TG signal field

and the probe pulse that are separated by $|T_w|$ (see newly added Figure 6f of the revised manuscript) – note that the fringe spacing along the y-axis is proportional to $1/|T_w|$. Figure S5 in the revised SI also clearly shows the relation between the fringe spacing of the spectral interferogram and time-domain signal, which is further confirmed by our simulation results with time-separated two Gaussian pulses. Second, the fringe patterns are roughly diagonally elongated in the 2DES spectra, which is well explained by a phase-shifted spectral interferogram scheme shown in (new) Figure 6g. Due to the inhomogeneous effective refractive indices of different sized gold nanorods, induced by various excitation frequencies, the TG signal fields at different excitation frequencies are slightly differently time-delayed from the probe pulse, giving rise to phase changes between the spectral interferograms for different excitation frequencies. Indeed, it is shown here that the fringe pattern provides information on the degree of inhomogeneity of gold nanorods with high sensitivity. To quantitatively describe this effect, we have newly performed 2D simulations of gold nanorods based on this phase-shifted scheme (Figures 6b and 6g) and compared them with the experimental data in the revised manuscript. In addition, detailed supplementary notes with illustrative schemes are added to the revised SI (pages 8-9).

Some minor observations:

4. They should include clearly what is the observable plotted. Currently all the 2D color maps and the captions are missing the information about what are the units and what is the observable.

Response: All the 2DES signals in the manuscript represent the differential transmitted spectra, $\Delta S(\omega_r, T_w, \omega_t) = S(\omega_r, \phi_{12}=0, T_w, \omega_t) - S(\omega_r, \phi_{12}=\pi, T_w, \omega_t)$, where ϕ_{12} is the relative phase shift between the replica pump pulses. This is known as the phase cycling scheme to measure the absorptive (imaginary part) signal of the third-order nonlinear susceptibility in a pump-probe two-beam geometry (Ref. 32). We have added a couple of sentences explaining what the observable is in the 2DES spectra (please see the revised figure caption of Figure 4 and the Method and materials section in page 14). Thanks.

5. On Page 9 they state: "Due to the coherent SPR effect of AuNRs, the preceding probe pulse induces an anomalously strong nonlinear electronic response, which considerably modulates the refractive index of AuNRs." For the reviewer this paragraph is not clear, they should make another effort at explaining what is the effect that is observed here. How does the coherent SPR effect induce a strong nonlinear response? I believe this is the probe so how can this be the case? What do they mean with anomalous? Which nonlinear electronic response is meant, an ultrafast coherent plasmonic polarisation within the plasmon dephasing time, or induced hot electron gas?

Response: As mentioned in our Response to the second comment of the reviewer 1, we have realized that the TG (multiple-pulse optical Kerr effect) is the main effect giving rise to the spectral interference in 2DES, not just the scattered pump by the preceding probe. We have added a new paragraph explaining this effect in the revised manuscript (new Figure 6f, main text in pages 10-11, and Supplementary notes in pages 8 and 9 in revised SI).

Here "a strong nonlinear electronic response" means the TG effect (signal) induced by strong coherent plasmonic polarizations of gold nanorods. When the non-collinearly propagating probe and first pump pulses interact with the gold nanorod sample, a spatial modulation of the refractive index (TG) is created by a spatially periodic excitation of these coherent plasmonic polarizations across the sample. Then the subsequent second pump pulse (pu2) is then diffracted by thus created TG to generate third-order TG signal field, which then produces a spectral interference along the probe axis after interfering with the preceding (by $|T_w|$) probe. What is interesting is that such effect

(spectral interference pattern) is notably strong for gold nanorods, while bio-organic systems such as zinc naphthalocyanine (Figure S3) and light-harvesting complex (Figure S4) do not show any pronounced oscillation (spectral interference and fringe pattern) in their 2DES spectra. Since the intensity of the diffracted TG signal depends on and decreases with the strength of the nonlinear effect involved, it is considered that a typical strength of the nonlinear effect of such bio-organic system is much weaker than what plasmonic metal nanoparticles have. This is why we stated that the observed effect results from “an anomalous strong nonlinear response” of plasmonic gold nanorods, which cannot be easily observed in other bio-organic materials.

6. A Conclusion paragraph is missing

Response: According to this comment, we have added a Conclusion paragraph (pages 13-14) in the revised manuscript. Thanks.

7. In general there is no effort at quantitative modelling of effects. Everything is a bit qualitative. Given the literature suggested in point 2., I would expect that they could provide some semi-quantitative description of the effect?

Response: To better explain the main effects (NLS and spectral interference) in the present study, we have performed extensive numerical simulations of 2DES spectra for the corresponding model cases (new Figure 2 for NLS and new Figure 6b for spectral interference) and tried to elucidate their physical origins and meanings by comparing them with the experimental results.

REVIEWERS' COMMENTS:

Reviewer #1 (Remarks to the Author):

The authors responded carefully and in detail to my earlier comments and made appropriate changes to the ms.

The added Figure 2 is a very nice illustration of how inhomogeneity affects the nodal line slope. My objection remains that the part of the paper dedicated to the strength of the short time transient signal is not extraordinary or unexpected. But, they have added an interesting discussion of how inhomogeneity can affect the temporal phase shift in the TG oscillations (Figure 6). A clear description of the methods and parameters used to simulate 2DES and TG spectra is needed in the "methods" section (some material may be moved from Figure Caption 2). The manuscript presentation has been somewhat improved, although the writing could be made more concise.

I believe the paper is now publishable though I remain uncertain as to the extent of its novelty.

Reviewer #2 (Remarks to the Author):

The authors have made significant modifications to the manuscript and have satisfactorily implemented all points raised. I am happy to recommend publication in Nature Communications in its current form.

Point-by-point responses to the Reviewers' comments and suggestions

Reviewer 1

Comment 1: *“The authors responded carefully and in detail to my earlier comments and made appropriate changes to the ms.*

The added Figure 2 is a very nice illustration of how inhomogeneity affects the nodal line slope. My objection remains that the part of the paper dedicated to the strength of the short time transient signal is not extraordinary or unexpected. But, they have added an interesting discussion of how inhomogeneity can affect the temporal phase shift in the TG oscillations (Figure 6). A clear description of the methods and parameters used to simulate 2DES and TG spectra is needed in the "methods" section (some material may be moved from Figure Caption 2). The manuscript presentation has been somewhat improved, although the writing could be made more concise.

Response: According to the reviewer's comment, we have added a paragraph (with the subheading **“Simulation of spectral interferometric 2DES spectra by TG”**) describing a detailed method for numerical simulations of the transient grating (TG) fringe pattern in the 2DES spectra in the “Methods” section (pages 14-15). We hope that the revised manuscript is now suitable for publication in Nature Communications.

Reviewer 2

Comment 1: *“The authors have made significant modifications to the manuscript and have satisfactorily implemented all points raised. I am happy to recommend publication in Nature Communications in its current form.”*

Response: We are very glad to know that the reviewer finds the submitted manuscript suitable for publication in Nature Communications.